# Rao-Blackwellizing the Straight-Through Gumbel-Softmax Gradient Estimator

**Max B. Paulus**
ETH Zürich
MPI for Intelligent Systems, Tübingen
max.paulus@inf.ethz.ch

**Chris J. Maddison**[*]
University of Toronto
Vector Institute
cmaddis@cs.toronto.edu

**Andreas Krause**
ETH Zürich
krausea@ethz.ch

## Abstract

Gradient estimation in models with discrete latent variables is a challenging problem, because the simplest unbiased estimators tend to have high variance. To counteract this, modern estimators either introduce bias, rely on multiple function evaluations, or use learned, input-dependent baselines. Thus, there is a need for estimators that require minimal tuning, are computationally cheap, and have low mean squared error. In this paper, we show that the variance of the straight-through variant of the popular Gumbel-Softmax estimator can be reduced through Rao-Blackwellization without increasing the number of function evaluations. This provably reduces the mean squared error. We empirically demonstrate that this leads to variance reduction, faster convergence, and generally improved performance in two unsupervised latent variable models.

## 1 Introduction

Models with discrete latent variables are common in machine learning. Discrete random variables provide an effective way to parameterize multi-modal distributions, and some domains naturally have latent discrete structure (e.g, parse trees in NLP). Thus, discrete latent variable models can be found across a diverse set of tasks, including conditional density estimation, generative text modelling (Yang et al., 2017), multi-agent reinforcement learning (Mordatch & Abbeel, 2017; Lowe et al., 2017) or conditional computation (Bengio et al., 2013; Davis & Arel, 2013).

The majority of these models are trained to minimize an expected loss using gradient-based optimization, so the problem of gradient estimation for discrete latent variable models has received considerable attention over recent years. Existing estimation techniques can be broadly categorized into two groups, based on whether they require one loss evaluation (Glynn, 1990; Williams, 1992; Bengio et al., 2013; Mnih & Gregor, 2014; Chung et al., 2017; Maddison et al., 2017; Jang et al., 2017; Grathwohl et al., 2018) or multiple loss evaluations (Gu et al., 2016; Mnih & Rezende, 2016; Tucker et al., 2017) per estimate. These estimators reduce variance by introducing bias or increasing the computational cost with the overall goal being to reduce the total mean squared error.

Because loss evaluations are costly in the modern deep learning age, single evaluation estimators are particularly desirable. This family of estimators can be further categorized into those that relax the discrete randomness in the forward pass of the model (Maddison et al., 2017; Jang et al., 2017; Paulus et al., 2020) and those that leave the loss computation unmodified (Glynn, 1990; Williams, 1992; Bengio et al., 2013; Chung et al., 2017; Mnih & Gregor, 2014; Grathwohl et al., 2018). The ones that do not modify the loss computation are preferred, because they avoid the accumulation of errors in the forward direction and they allow the model to exploit the sparsity of discrete computation. Thus, there is a particular need for single evaluation estimators that do not modify the loss computation.

---

[*]Work done partly at the Institute for Advanced Study, Princeton, NJ.

In this paper we introduce such a method. In particular, we propose a Rao-Blackwellization scheme for the straight-through variant of the Gumbel-Softmax estimator (Jang et al., 2017; Maddison et al., 2017), which comes at a minimal cost, and does not increase the number of function evaluations. The *straight-through Gumbel-Softmax estimator* (ST-GS, Jang et al., 2017) is a lightweight state-of-the-art single-evaluation estimator based on the Gumbel-Max trick (see Maddison et al., 2014, and references therein). The ST-GS uses the argmax over Gumbel random variables to generate a discrete random outcome in the forward pass. It computes derivatives via backpropagation through a tempered *softmax* of the same Gumbel sample. Our Rao-Blackwellization scheme is based on the key insight that there are *many* configurations of Gumbels corresponding to the *same* discrete random outcome and that these can be marginalized over with Monte Carlo estimation. By design, there is no need to re-evaluate the loss and the additional cost of our estimator is linear only in the number of Gumbels needed for a single forward pass. As we show, the Rao-Blackwell theorem implies that our estimator has lower mean squared error than the vanilla ST-GS. We demonstrate the effectiveness of our estimator in unsupervised parsing on the ListOps dataset (Nangia & Bowman, 2018) and on a variational autoencoder loss (Kingma & Welling, 2013; Rezende et al., 2014). We find that in practice our estimator trains *faster* and achieves *better test set performance*. The magnitude of the improvement depends on several factors, but is particularly pronounced at small batch sizes and low temperatures.

## 2 BACKGROUND

For clarity, we consider the following simplified scenario. Let $D \sim p_\theta$ be a discrete random variable $D \in \{0,1\}^n$ in a one-hot encoding, $\sum D_i = 1$, with distribution given by $p_\theta(D) \propto \exp(D^T\theta)$ where $\theta \in \mathbb{R}^n$. Given a continuously differentiable $f : \mathbb{R}^{2n} \to \mathbb{R}$, we wish to minimize,

$$\min_\theta \mathbb{E}[f(D,\theta)], \tag{1}$$

where the expectation is taken over all of the randomness. In general $\theta$ may be computed with some neural network, so our aim is to derive estimators of the total derivative of the expectation with respect to $\theta$ for use in stochastic gradient descent. This framework covers most simple discrete latent variable models, including variational autoencoders (Kingma & Welling, 2013; Rezende et al., 2014).

The *REINFORCE estimator* (Glynn, 1990; Williams, 1992) is unbiased (under certain smoothness assumptions) and given by:

$$\nabla_{\text{REINF}} := f(D,\theta)\frac{\partial \log p_\theta(D)}{\partial \theta} + \frac{\partial f(D,\theta)}{\partial \theta}. \tag{2}$$

Without careful use of control variates (Mnih & Gregor, 2014; Tucker et al., 2017; Grathwohl et al., 2018), the REINFORCE estimator tends to have prohibitively high variance. To simplify exposition we assume henceforth that $f(D,\theta) = f(D)$ does not depend on $\theta$, because the dependence of $f(D,\theta)$ on $\theta$ is accounted for in the second term of (2), which is shared by most estimators and generally has low variance.

One strategy for reducing the variance is to introduce bias through a relaxation (Jang et al., 2017; Maddison et al., 2017). Define the tempered softmax $\text{softmax}_\tau : \mathbb{R}^n \to \mathbb{R}^n$ by $\text{softmax}_\tau(x)_i = \exp(x_i/\tau)/\sum_{j=1}^n \exp(x_j/\tau)$. The relaxations are based on the observation that the sampling of $D$ can be reparameterized using Gumbel random variables and the zero-temperature limit of the tempered softmax under the coupling:

$$D = \lim_{\tau \to 0} S_\tau; \qquad S_\tau = \text{softmax}_\tau(\theta + G) \tag{3}$$

where $G$ is a vector of i.i.d. $G_i \sim \text{Gumbel}$ random variables. At finite temperatures $S_\tau$ is known as a Gumbel-Softmax (GS) (Jang et al., 2017) or concrete (Maddison et al., 2017) random variable, and the relaxed loss $\mathbb{E}[f(S_\tau,\theta)]$ admits the following reparameterization gradient estimator for $\tau > 0$:[1]

$$\nabla_{\text{GS}} := \frac{\partial f(S_\tau)}{\partial S_\tau}\frac{d\,\text{softmax}_\tau(\theta + G)}{d\theta}. \tag{4}$$

---

[1] For a function $f(x_1, x_2)$, $\partial f(z_1, z_2)/\partial x_1$ is the partial derivative (e.g., a gradient vector) of $f$ in the first variable evaluated at $z_1, z_2$. For a function $g(\theta)$, $dg/d\theta$ is the total derivative of $g$ in $\theta$. For example, $d\,\text{softmax}_\tau(\theta + G)/d\theta$ is the Jacobian of the tempered softmax evaluated at the random variable $\theta + G$.

This is an unbiased estimator of the gradient of $\mathbb{E}[f(S_\tau, \theta)]$, but a biased estimator of our original problem (1). For this to be well-defined $f$ must be defined on the interior of the simplex (where $S_\tau$ sits). This estimator has the advantage that it is easy to implement and generally low-variance, but the disadvantage that it modifies the forward computation of $f$ and is biased. Henceforth, we assume $D, S_\tau$, and $G$ are coupled almost surely through (3).

Another popular family of estimators are the so-called *straight-through estimators* (c.f., Bengio et al., 2013; Chung et al., 2017). In this family, the forward computation of $f$ is unchanged, but backpropagation is computed "through" a surrogate. One popular variant takes as a surrogate the tempered probabilities of $D$, resulting in the *slope-annealed straight-through estimator (ST)*:

$$\nabla_{\text{ST}} := \frac{\partial f(D)}{\partial D} \frac{d \, \text{softmax}_\tau(\theta)}{d\theta}. \tag{5}$$

For binary $D$, a lower bias variant of this estimator (FouST) was proposed in Pervez et al. (2020).

The most popular straight-through estimator is known as the *straight-through Gumbel-Softmax* (ST-GS, Jang et al., 2017). The surrogate for ST-GS is $S_\tau$, whose Gumbels are coupled to $D$ through (3):

$$\nabla_{\text{STGS}} := \frac{\partial f(D)}{\partial D} \frac{d \, \text{softmax}_\tau(\theta + G)}{d\theta}. \tag{6}$$

The straight-through family has the advantage that they tend to be low-variance and $f$ need not be defined on the interior of the simplex (although $f$ must be differentiable at the corners). This family has the disadvantage that they are not known to be unbiased estimators of *any* gradient. These estimators are quite popular in practice, because they preserve the forward computation of $f$, which prevents the forward propagation of errors and maintains sparsity (Choi et al., 2017; Chung et al., 2017; Bengio et al., 2013).

All of the estimators discussed in this paper can be computed by any of the standard automatic differentiation software packages using a single evaluation of $f$ on a realization of $D$ or some underlying randomness. We present implementation details for these and our Gumbel-Rao estimator in the Appendix, emphasizing the surrogate loss framework (Schulman et al., 2015; Weber et al., 2019) and considering the multiple stochastic layer case not covered by (1).

## 3 GUMBEL-RAO GRADIENT ESTIMATOR

### 3.1 RAO-BLACKWELLIZATION OF ST-GUMBEL-SOFTMAX

We now derive our Rao-Blackwelization scheme for the ST-GS estimator. Our approach is based on the observation that there is a *many-to-one* relationship between realizations of $\theta + G$ and $D$ in the coupling described by (3) and that the variance introduced by $\theta + G$ can be marginalized out. The resulting estimator, which we call the *Gumbel-Rao (GR)* estimator, is guaranteed by the Rao-Blackwell theorem to have lower variance than ST-GS. In the next subsection we turn to the practical question of carrying out this marginalization.

In the Gumbel-max trick (3), $D$ is a one-hot indicator of the index of $\arg\max_i \{\theta_i + G_i\}$. Because this argmax operation is non-invertible, there are many configurations of $\theta + G$ that correspond to a single $D$ outcome. Consider an alternate factorization of the joint distribution of $(\theta + G, D)$: first sample $D \sim p_\theta$, and then $\theta + G$ given $D$. In this view, the Gumbels are auxillary random variables, at which the Jacobian of the tempered softmax is evaluated and which locally increase the variance of the estimator. This local variance can be removed by marginalization. This is the key insight of our GR estimator, which is given by,

$$\nabla_{\text{GR}} := \frac{\partial f(D)}{\partial D} \mathbb{E}\left[\frac{d \, \text{softmax}_\tau(\theta + G)}{d\theta}\middle| D\right]. \tag{7}$$

It is not too difficult to see that $\nabla_{\text{GR}} = \mathbb{E}\left[\nabla_{\text{STGS}}|D\right]$. By the tower rule of expectation, GR has the same expected value as ST-GS and is an instance of a Rao-Blackwell estimator (Blackwell, 1947; Rao, 1992). Thus, it has the same mean as ST-GS, but a lower variance. Taken together, these facts imply that GR enjoys a lower mean squared error (*not* a lower bias) than ST-GS.

**Proposition 1.** Let $\nabla_{\text{STGS}}$ and $\nabla_{\text{GR}}$ be the estimators defined in (6) and (7). Let $\nabla_\theta := d\mathbb{E}[f(D)]/d\theta$ be the true gradient that we are trying to estimate. We have

$$\mathbb{E}\left[\|\nabla_{\text{GR}} - \nabla_\theta\|^2\right] \leq \mathbb{E}\left[\|\nabla_{\text{STGS}} - \nabla_\theta\|^2\right]. \tag{8}$$

*Proof.* The proposition follows from Jensen's inequality and the linearity of expectations, see C.1. $\square$

While GR is only guaranteed to reduce the variance of ST-GS, Proposition 1 guarantees that, as a function of $\tau$, the MSE of GR is a pointwise lower bound on ST-GS. This means GR can be used for estimation at temperatures, where ST-GS has low bias but prohibitively high variance. Thus, GR extends the region of suitable temperatures over which one can tune. This allows a practitioner to explore an expanded set when trading-off of bias and variance. Empirically, lower temperatures tend to reduce the bias of ST-GS, but we are not aware of any work that studies the convergence of the derivative in the temperature limit. In our experiments, we observe that our estimator facilitates training at lower temperatures to improve in both bias and variance over ST-GS. Thus, our estimator retains the favourable properties of ST-GS (single, unmodified evaluation of $f$) while improving its performance.

## 3.2 MONTE CARLO APPROXIMATION

The GR estimator requires computing the expected value of the Jacobian of the tempered softmax over the distribution $\theta + G|D$. Unfortunately, an analytical expression for this is only available in the simplest cases.[2] In this section we provide a simple Monte Carlo (MC) estimator with sample size $K$ for $\mathbb{E}[dS_\tau/d\theta|D]$, which we call the *Gumbel-Rao Monte Carlo Estimator (GR-MCK)*. This estimator can be computed locally at a cost that only scales like $nK$ (the arity of $D$ times $K$).

They key property exploited by GR-MC$K$ is that $\theta + G|D$ can be reparameterized in the following closed form. Given a realization of $D$ such that $D_i = 1$, $Z(\theta) = \sum_{i=1}^n \exp(\theta_i)$, and $E_j \sim$ exponential i.i.d., we have the following equivalence in distribution (Maddison et al., 2014; Maddison, 2016; Tucker et al., 2017).

$$\theta_j + G_j|D \stackrel{d}{=} \begin{cases} -\log(E_j) + \log Z(\theta) & \text{if } j = i \\ -\log\left(\frac{E_j}{\exp(\theta_j)} + \frac{E_i}{Z(\theta)}\right) & \text{o.w.} \end{cases} \tag{9}$$

With this in mind, we define the GR-MC$K$ estimator:

$$\nabla_{\text{GRMC}K} := \frac{\partial f(D)}{\partial D}\left[\frac{1}{K}\sum_{k=1}^K \frac{d\,\text{softmax}_\tau(G_\theta^k)}{d\theta}\right], \tag{10}$$

where $G_\theta^k \sim \theta + G|D$ i.i.d. using the reparameterization (9). For the case $K = 1$, our estimator reduces to the standard ST-GS estimator. The cost for drawing multiple samples $G_\theta^k \sim \theta + G|D$ scales only *linearly* in the arity of $D$ and is usually negligible in modern applications, where the bulk of computation accrues from the computation of $f$. Moreover, drawing multiple samples of $\theta + G|D$ can easily be parallelized on modern workstations (GPUs, etc.). Our estimator remains a single-evaluation estimator under this scheme, because the loss function $f$ is still only evaluated at $D$. Finally, as with GR, the GR-MC$K$ is guaranteed to improve in MSE over ST-GS for any $K \geq 1$, as confirmed in Proposition 2.

**Proposition 2.** Let $\nabla_{\text{STGS}}$ and $\nabla_{\text{GRMC}K}$ be the estimators defined in (6) and (10). Let $\nabla_\theta := d\mathbb{E}[f(D)]/d\theta$ be the true gradient that we are trying to estimate. For all $K \geq 1$, we have

$$\mathbb{E}\left[\|\nabla_{\text{GRMC}K} - \nabla_\theta\|^2\right] \leq \mathbb{E}\left[\|\nabla_{\text{STGS}} - \nabla_\theta\|^2\right]. \tag{11}$$

*Proof.* The proposition follows from Jensen's inequality and the linearity of expectations, see C.2. $\square$

---

[2]For example, in the case of $n = 2$ (binary) and $\tau = 1$ an analytical expression for the GR estimator is available.

### 3.3 Variance Reduction in Minibatches

The variance of GR-MC$K$ can be reduced by increasing $K$ or by averaging $B$ i.i.d. samples of the GR-MC$K$ estimator. An average of i.i.d. samples $\nabla^b_{\mathrm{GRMC}K}$ for $b \in \{1, \ldots, B\}$ is a generalization of minibatching by sampling data points with replacement. In particular, $\theta$ may depend on an additional source of randomness, i.e., $\theta = h(X)$ for $X \sim \mathcal{P}$. In this case, $\nabla_{\mathrm{STGS}}$ is a random variable that depends not only on $D$ and $G$, but also on $X$. In this subsection, we consider the effect of increasing $K$ and $B$ separately. Expectations are taken over all the randomness.

Let $\nabla^b_{\mathrm{GRMC}K}$ be i.i.d. as $\nabla_{\mathrm{GRMC}K}$ for $b \in \{1, \ldots, B\}$ and define the following "minibatched" GR-MC$K$ estimator:

$$\overline{\nabla}^{1:B}_{\mathrm{GRMC}K} := \frac{1}{B} \sum_{b=1}^{B} \nabla^b_{\mathrm{GRMC}K}. \tag{12}$$

Proposition 3 summarizes the scaling of the variance of (12), and is an elementary application of the law of total variance.

**Proposition 3.** Let $\nabla_{\mathrm{STGS}}$, $\nabla_{\mathrm{GR}}$ and $\overline{\nabla}^{1:B}_{\mathrm{GRMC}K}$ be the estimators defined in (6), (7) and (12). We have

$$\mathrm{var}\left[\overline{\nabla}^{1:B}_{\mathrm{GRMC}K}\right] = \frac{\mathbb{E}\left[\mathrm{var}\left[\nabla_{\mathrm{STGS}}|D, X\right]\right]}{BK} + \frac{\mathrm{var}\left[\nabla_{\mathrm{GR}}\right]}{B} \tag{13}$$

where $\mathrm{var}$ is the trace of the covariance matrix.

*Proof.* The proposition follows directly from the law of total variance, see C.3. ☐

As expected the total variance of $\overline{\nabla}^{1:B}_{\mathrm{GRMC}K}$ decreases like $1/B$. The key point of Proposition 3 is that the component of the variance that $K$ reduces can also be reduced by increasing the batch size $B$. This suggests that the effect of GR-MC$K$ will be most pronounced at small batch sizes. Proposition 3 also indicates that there are diminishing returns to increasing $K$ for a fixed batch size $B$, such that the variance of GR-MC$K$ will eventually be dominated by the right-hand term of (13). In our experimental section, we explore various $K$ and study the effect on gradient estimation in more detail.

Finally, we note that the choice of a Monte Carlo scheme to approximate $\mathbb{E}\left[dS_\tau/d\theta|D\right]$ permits the use of additional well-known variance reduction methods to improve the estimation properties of our gradient estimator. For example, antithetic variates or importance sampling are sensible methods to explore in this setting (Kroese et al., 2013). For low-dimensional discrete random variables, Gaussian quadrature or other numerical methods could be employed. However, we found the simple Monte Carlo scheme described above effective in practice and report results based on this procedure in the experimental section.

## 4 Related Work

The idea of using Rao-Blackwellization to reduce the variance of gradient estimators for discrete latent variable models has been explored in machine learning. For example, Liu et al. (2018) describe a sum-and-sample style estimator that analytically computes part of the expectation to reduce the variance of the gradient estimates. The favorable properties of their estimator are due to the Rao-Blackwell theorem. Kool et al. (2020) describe a gradient estimator based on sampling without replacement. Their estimator emerges naturally as the Rao-Blackwell estimator of the importance-weighted estimator (Vieira, 2017) and the estimator described by Liu et al. (2018). Both of these estimators rely on *multiple* function evaluations to compute a gradient estimate. In contrast, our work is the first to consider Rao-Blackwellisation in the context of a *single-evaluation* estimator.

Recently, Paulus et al. (2020) extend the Gumbel-Softmax gradient estimator to other discrete structures. Our approach can be used to reduce the variance of the corresponding straight-through variants, when an efficient reparameterization of the perturbation conditional on the discrete structure is available (Gane et al., 2014).

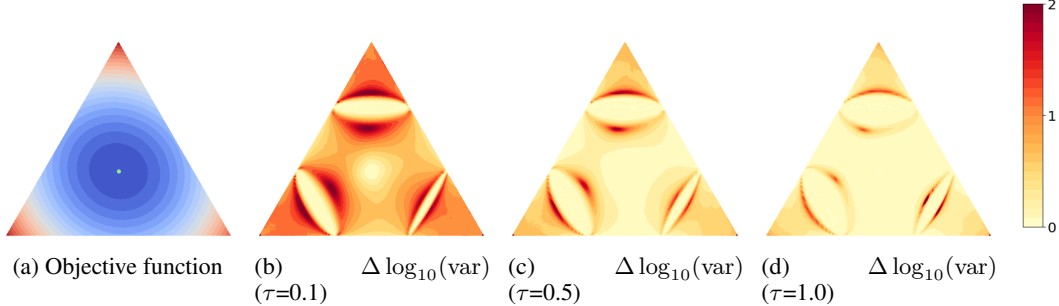

(a) Objective function    (b) $\quad \Delta \log_{10}(\text{var})$    (c) $\quad \Delta \log_{10}(\text{var})$    (d) $\quad \Delta \log_{10}(\text{var})$
$(\tau=0.1)$                $(\tau=0.5)$                $(\tau=1.0)$

Figure 1: Our estimator (GR-MC$K$) effectively reduces the variance over the entire simplex and is particularly effective at low temperatures. Contours for the quadratic programme in three dimensions (1a) and difference in log10-trace of the covariance matrix between ST-GS and GR-MC1000 at different temperatures (1b, 1c, 1d). Warmer means difference is larger.

## 5 EXPERIMENTS

### 5.1 PROTOCOL

In this section, we study the effectiveness of our gradient estimator in practice. In particular, we evaluate its performance with respect to the temperature $\tau$, the number of MC samples $K$ and the batch size $B$. We measure the variance reduction and improvements in MSE our estimator achieves in practice, and assess whether its lower variance gradient estimates accelerate the convergence on the objective or improve final test set performance. Our focus is on single-evaluation gradient estimation and we compare against other non-relaxing estimators (ST, FouST, ST-GS and REINFORCE with a running mean as a baseline) and relaxing estimators (GS), where permissible. Experimental details are given in Appendix D.

First, we consider a toy example which allows us to explore and visualize the variance of our estimator and suggests that it is particularly effective at low temperatures. Next, we evaluate the effect of $\tau$ and $K$ in a latent parse tree task which does not permit the use of relaxed gradient estimators. Here, our estimator facilitates training at low temperatures to improve overall performance and is effective even with few MC samples. Finally, we train variational auto-encoders with discrete latent variables (Kingma & Welling, 2013; Rezende et al., 2014). Our estimator yields improvements at small batch sizes and obtains competitive or better performance than the GS estimator at the largest arity.

### 5.2 QUADRATIC PROGRAMMING ON THE SIMPLEX

As a toy problem, we consider the problem of minimizing a quadratic program $(p-c)^\intercal Q(p-c)$ over the probability simplex $\Delta^{n-1} = \{p \in R^n : p_i \geq 0, \sum_{i=1}^n p_i = 1\}$ for $Q \in R^{n \times n}$ positive-definite and $c \in R^n$. This problem may be reframed as the following stochastic optimization problem,

$$\min_{p \in \Delta^{n-1}} \mathbb{E}[(D-c)^\intercal A(p)(D-c)],$$

where $D \sim \text{Discrete}(p)$ and $A_{ii}(p) = \frac{(p_i-c_i)^2}{p_i-2p_ic_i+c_i^2}Q_{ii}$ and $A_{ij}(p) = \frac{(p_i-c_i)(p_j-c_j)}{c_ic_j-p_ic_j-c_ip_j}Q_{ij}$ for $i \neq j$. While solving the above problem is simple using standard methods, it provides a useful testbed to evaluate the effectiveness of our variance reduction scheme. For this purpose, we consider $Q_{ij} = \exp(-2|i-j|)$ and $c_i = \frac{1}{3}$ in three dimensions.

Our estimator reduces the variance in the gradient estimation over the entire simplex and is particularly effective at low temperatures in this problem. In Figure 1, we compare the log10-trace of the covariance matrix of ST-GS and GR-MC1000 at three different temperatures and display their difference over the entire domain. The improvement is universal. The pattern is not always intuitive (oval bull's eyes), despite the simplicity of the objective function. Compared with ST-GS, our estimator on this example appears more effective closer to the corners and edges, which is important for learning discrete distributions. At lower temperatures, the difference between the two estimators becomes particularly acute. This suggests that our estimator may train better at lower temperatures and be more responsive to optimizing over the temperature to successfully trade off bias and variance.

Table 1: Our estimator (GR-MC$K$) facilitates training at lower temperatures with improved performance on the latent parse tree task. Best test classification accuracy on the ListOps dataset selected on the validation set. Best estimator at given temperature in bold, best estimator across temperatures in italics. Higher is better.

| | $L \leq 10$ | | | $L \leq 25$ | | | $L \leq 50$ | | |
|---|---|---|---|---|---|---|---|---|---|
| ESTIMATOR | $\tau = 0.01$ | $\tau = 0.1$ | $\tau = 1.0$ | $\tau = 0.01$ | $\tau = 0.1$ | $\tau = 1.0$ | $\tau = 0.01$ | $\tau = 0.1$ | $\tau = 1.0$ |
| ST-GS | 38.8 | 59.3 | 65.8 | 41.2 | 57.1 | 60.2 | 46.8 | 56.8 | 59.6 |
| GR-MC10 | 66.4 | 66.9 | 66.7 | 60.7 | 60.8 | 60.9 | 58.7 | 59.1 | 59.6 |
| GR-MC100 | 65.6 | 66.3 | 65.9 | 60.0 | *61.3* | **61.2** | 59.6 | 59.1 | 59.6 |
| GR-MC1000 | **66.5** | *67.1* | **67.0** | **60.2** | 60.9 | **61.2** | *60.0* | **59.8** | **59.9** |

## 5.3 UNSUPERVISED PARSING ON LISTOPS

Straight-through estimators feature prominently in NLP (Martins et al., 2019) where latent discrete structure arises naturally, but the use of relaxations is often infeasible. Therefore, we evaluate our estimator in a latent parse tree task on subsets of the ListOps dataset (Nangia & Bowman, 2018). This dataset contains sequences of prefix arithmetic expressions $x$ (e.g., `max[ 3 min[ 8 2 ]]`) that evaluate to an integer $y \in \{0, 1, \ldots 9\}$. The arithmetic syntax induces a latent parse tree $T$. We consider the model by (Choi et al., 2017) that learns a distribution over plausible parse trees of a given sequence to maximize

$$\mathbb{E}_{q_\theta(T|x)} \left[ \log p_\phi(y|T, x) \right].$$

Both the conditional distribution over parse trees $q_\theta(T|x)$ and the classifier $p_\phi(y|T, x)$ are parameterized using neural networks. In this model, a parse tree $T \sim q_\theta(T|x)$ for a given sentence is sampled bottom-up by successively combining the embeddings of two tokens that appear in a given sequence until a single embedding for the entire sequence remains. This is then used for performing the subsequent classification. Because it is computationally infeasible to marginalize over all trees, Choi et al. (2017) rely on the ST-GS estimator for training. We compare this estimator against our estimator GR-MC$K$ with $K \in \{10, 100, 1000\}$. We consider temperatures $\tau \in \{0.01, 0.1, 1.0\}$ and experiment with shallow and deeper trees by considering sequences of length $L$ up to 10, 25 and 50. All models are trained with stochastic gradient descent with a batch size equal to the maximum $L$. Because we are interested in a controlled setting to investigate the effect of $\tau$ and $K$, our experimental set-up is significantly simpler than elsewhere (e.g., Havrylov et al., 2019). We give details and highlight important differences in Appendix D.1.

Our estimator facilitates training at lower temperatures and achieves better final test set accuracy than ST-GS (Table 1). Increasing $K$ improves the performance at low temperatures, where the differences between the estimators are most pronounced. Overall, across all temperatures this results in modest improvements, particularly for shallow trees and small batch sizes. We also find evidence for diminishing returns: The differences between ST-GS and GR-MC10 are larger than between GR-MC100 or GR-MC1000, suggesting that our estimator is effective even with few MC samples.

## 5.4 GENERATIVE MODELING WITH DISCRETE VARIATIONAL AUTO-ENCODERS

Finally, we train variational auto-encoders (Kingma & Welling, 2013; Rezende et al., 2014) with discrete latent random variables on the MNIST dataset of handwritten digits (LeCun & Cortes, 2010). We used the fixed binarization of (Salakhutdinov & Murray, 2008) and the standard split into train, validation and test sets. Our objective is to maximize the following variational lower bound on the log-likelihood,

$$\log p(x) > \mathbb{E}_{q_\theta(D^i|x)} \left[ \log \left( \frac{1}{M} \sum_{j=1}^{M} \frac{p_\phi(x, D^i)}{q_\theta(D^j|x)} \right) \right]$$

where $x$ denotes the input image and $D^i \sim q_\theta(D^i|x)$ denotes a vector of discrete latent random variables. This objective takes a form in equation (1). For training, the bound is approximated using

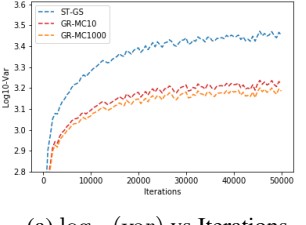 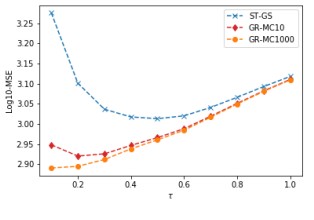 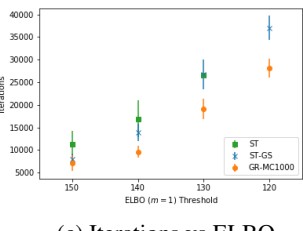

(a) $\log_{10}(\text{var})$ vs Iterations  (b) $\log_{10}(\text{mse})$ vs $\tau$  (c) Iterations vs ELBO

Figure 2: Our estimator (GR-MC$K$) effectively reduces the variance over the entire training trajectory (2a), achieves a lower mean squared error at a lower temperature (2b) and converges faster than ST and ST-GS on the discrete VAE objective (2c). Log10-trace of the covariance matrix over a training trajectory (2a) and log10-MSE (2b) at different temperatures during training, average number of iterations and standard error to reach various thresholds of the objective on the validation set (2c).

only a single sample ($M = 1$). For final validation and testing, we use 5000 samples ($M = 5000$). Both the generative model $p_\phi(x, D)$ and the variational distributions $q_\theta(D|x)$ were parameterized using neural networks. We experiment with different batch sizes and discrete random variables of arities in $\{2, 4, 8, 16\}$ as in Maddison et al. (2017). To facilitate comparisons, we do not alter the total dimension of the latent space and train all models for 50,000 iterations using stochastic gradient descent with momentum. Hyperparameters are optimised for each estimator using random search (Bergstra & Bengio, 2012) over twenty independent runs. More details are given in Appendix D.2.

Our estimator effectively reduces the variance over the entire training trajectory (Figure 2a). Even a small number of MC samples ($K = 10$) results in sizable variance reductions. The variance reduction compares favorably to the magnitude of the minibatch variance (Appendix E). Empirically, we find that lower temperatures tend to reduce bias. Our estimator facilitates training at lower temperatures and thus features a lower MSE (Figure 2b). During training our estimator can trade off bias and variance to improve the gradient estimation. Empirically, we observed that on this task, the best models using ST-GS trained at an average temperature of $0.65$, while the best models using GR-MC1000 trained at an average temperature of $0.35$. This is interesting, because it indicates that our estimator may make the use of temperature annealing during training more effective. We find lower variance gradient estimates improve convergence of the objective (Figure 2c). GR-MC1000 reaches various performance thresholds on the validation set with reliably fewer iterations than ST or ST-GS. This effect is observable at different arities and persistent over the entire training trajectory.

For final test set performance, our estimator outperforms REINFORCE and all other straight-through estimators (Table 2). The improvements over ST-GS extend up to two nats (for batch size 20, 16-ary) at small batch sizes and are more modest at large batch sizes as expected (also see Appendix E). This confirms that our estimator might be particularly effective in settings, where training at high batch sizes is prohibitively expensive. The improvements from increasing the number of MC samples tend to saturate at $K = 100$ on this task. Further, our results suggest that relaxed estimators may be preferred (if they can be used) for discrete random variables of smaller arity. For example, the GS estimator outperforms all straight-through estimators for binary variables for both batch sizes. For large arities however, we find that straight-through estimators can perform competitively: Our estimator GR-MC1000 achieves the best performance overall and outperforms the GS estimator for 16-ary variables.

## 6  CONCLUSION

We introduced the Gumbel-Rao estimator, a new single-evaluation non-relaxing gradient estimator for models with discrete random variables. Our estimator is a Rao-Blackwellization of the state-of-the-art straight-through Gumbel-Softmax estimator. It enjoys lower variance and can be implemented efficiently using Monte Carlo methods. In particular and in contrast to most other work, it does not require additional function evaluations. Empirically, our estimator improved final test set performance in an unsupervised parsing task and on a variational auto-encoder loss. It accelerated convergence on the objective and compared favorably to other standard gradient estimators. Even though the

Table 2: Our estimator (GR-MC$K$) outperforms other straight-through estimators for discrete-latent-space VAE objectives on the MNIST dataset and is competitive with the Gumbel-Softmax ($GS$) at large arities. Best bound on the test negative log-likelihood selected on the validation set. Best straight-through estimator in bold, best estimator in italics. Lower is better.

| | BINARY | | 4-ARY | | 8-ARY | | 16-ARY | |
|---|---|---|---|---|---|---|---|---|
| ESTIMATOR | $B = 20$ | $B = 200$ | $B = 20$ | $B = 200$ | $B = 20$ | $B = 200$ | $B = 20$ | $B = 200$ |
| GS | *98.2* | *96.4* | *95.7* | *93.8* | *95.5* | *92.3* | *96.8* | *94.3* |
| REINFORCE | 202.6 | 121.4 | 173.7 | 122.2 | 203.9 | 124.9 | 169.4 | 129.5 |
| ST | 105.5 | 103.1 | 106.2 | 104.5 | 107.2 | 105.1 | 108.2 | 104.5 |
| FOUST | 101.5 | 97.8 | - | - | - | - | - | - |
| ST-GS | 100.7 | 97.1 | 99.1 | 93.7 | 98.0 | 92.8 | 98.8 | 92.6 |
| GR-MC10 | 100.7 | 97.4 | 97.8 | 93.8 | 97.4 | 93.1 | 97.9 | 92.4 |
| GR-MC100 | 100.6 | **96.8** | **97.5** | 94.0 | 96.8 | *92.2* | 97.3 | 92.4 |
| GR-MC1000 | **100.5** | 97.0 | 97.6 | *93.5* | **96.5** | 92.5 | *96.8* | *92.2* |

gains were sometimes modest, they were persistent and particularly pronounced when models must be trained at low temperatures or with small batch sizes. We expect that our estimator will be most effective in such settings and that further gains may be uncovered when combining our Rao-Blackwellisation scheme with an annealing schedule for the temperature. Finally, we hope that our work inspires further exploration of the use of Rao-Blackwellisation for gradient estimation.

## ACKNOWLEDGEMENTS

MBP gratefully acknowledges support from the Max Planck ETH Center for Learning Systems. CJM is grateful for the support of the James D. Wolfensohn Fund at the Institute of Advanced Studies in Princeton, NJ. Resources used in preparing this research were provided, in part, by the Sustainable Chemical Processes through Catalysis (Suchcat) National Center of Competence in Research (NCCR), the Province of Ontario, the Government of Canada through CIFAR, and companies sponsoring the Vector Institute.

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
