# OpenReview forum: "Rao-Blackwellizing the Straight-Through Gumbel-Softmax Gradient Estimator"
_ICLR.cc/2021/Conference — ICLR 2021 Oral_

### Official Review · AnonReviewer4 · 2020-10-26
**Good work**

**Rating:** 8
**Confidence:** 3

**Review:**

Summary: The paper presents a new way algorithm to compute the straight-through variant of the Gumbel Softmax gradient estimator. The method does not change the estimator's bias, but provably reduces its variance (with a small overhead, using Rao-blackwellization). The new estimator shows good performance on different tasks, and appears to lead to more efficient optimization for lower temperatures (lower bias).

Clarity: The paper is well written.

Originality: The use of Rao-blackwellization in the proposed way is, up to the best of my knowledge, novel.

Pros of the paper and significance:
- Relaxation-based gradient estimators are widely used, and the proposed method may their variance quite significantly.
- The proposed algorithm has a clear justification from a theoretical perspective, and admits a simple implementation.
- The proposed algorithm does not require additional model evaluations, and thus may lead to large reductions in variance without incurring a high computational cost.
- The proposed method leads to more efficient optimization at lower temperatures (lower temperature translates to lower bias, but often higher variances).

Cons: I'd say one thing that could be included are additional baselines in the experimental section. There are other estimators that may be use. For instance, you could compare against VIMCO. While this is a different type of estimator (non single evaluation, not based on relaxations), it could be interesting to see how the results compare using this estimator too.

Recommendation: Accept (reasons in the "pros" list above).

---

> ### Author Response · Authors · 2020-11-18
> **We implemented an additional baseline.**
>
> **Thank you very much for your review and the positive reception of our work.**
>
> We have addressed your concern and incorporated an additional single-evaluation baseline, i.e., Pervez et al. (2020), based on the suggestion of R1. The reason we do not compare to multi-evaluation estimators (e.g. VIMCO) is that is very difficult to provide a level playing field: Each training iteration is significantly more expensive and even when basing comparisons on training time instead, these estimators may require significantly more memory, possibly training on multiple GPUs, which confounds comparisons.

---

### Official Review · AnonReviewer2 · 2020-10-28
**cheap and effective variance reduction trick to improve a widely adopted gradient estimator (for discrete distributions)**

**Rating:** 7
**Confidence:** 4

**Review:**



This paper introduces the Rao-blackwellization technique to reduce the variance of the straight-through gumbel-softmax gradient (STGS) estimator wrt the parameters of discrete distributions. The proposed method introduces almost trivial computational costs (relative to function evaluations) and is empirically and theoretically shown to systematically improve STGS.

I don’t have a lot of nitpicking to make for this paper, as it is quite well executed. The proposed method is very clean and the improvement over the STGS baseline is very consistent, and makes it even competitive with concrete-relaxation in the discrete latent variable model experiment.

Details:

Why not show the curve of ELBO during training, but the arrival-time-at-certain-thresholds in Fig 2-c?
Last paragraph of sec 5.4: The larger batch size here also reduces the variance of minibatch SGD, not just the variance of $\nabla_{GR}$ in (13). In fact each instance is a different approximate posterior, which has different base variance. This makes the discussion in 3.3 a bit misleading.

Suggestions:
For figure 1, perhaps visualize the variance of both separately. An improvement by 2 is not that meaningful if the variances of both are >> 2.

-- After rebuttal

Thank you for revising the paper. I've read the revised section, and stand by my original evaluation.

---

> ### Author Response · Authors · 2020-11-18
> **We clarified Section 3.3.**
>
> **Thank you very much for your review and the positive reception of our work.**
>
> We apologize for any confusion the discussion in 3.3 may have caused. We have addressed your concern and clarified this point in the revision by adding a footnote: Indeed, each instance has a different approximate posterior. GR estimates “joint parameters” that parameterize this approximate posterior distribution. The dataset is assumed i.i.d and the expectation over the data is omitted to improve readability.

---

### Official Review · AnonReviewer1 · 2020-10-29
**Method allows training of gumbel straight-through at lower temperatures, but lower temperature gains seem small**

**Rating:** 7
**Confidence:** 3

**Review:**

Summary:
* This paper proposes a Rao-Blackwellized version of the straight-through gumbel-softmax gradient (STGS) estimator.
* The Gumbel-Rao estimator remains single-evaluation (but multiple sample), does not have higher variance than the original straight-through estimator.
* The estimator exhibits lower variance at lower temperatures in the experiments.

Contributions:
* Proposes a single-evaluation estimator that cannot have higher variance than the STGS gradient estimator.
* Demonstrates effectiveness of proposed estimator in terms of the variance of the gradient estimator and the ELBO on a toy task, a simple parsing task (ListOps), and a mixture model for MNIST.

Strengths:
* The method is simple and the computational overhead is very small compared to the original STGS estimator.
* The empirical results support lower variance claims and effectiveness at lower temperature.

Weaknesses:
* I am not convinced that the relative gains from training at lower temperatures are significant.
* The overall gains over ST-GS seem to be modest on MNIST as well as the L <= 50 setting in ListOps.
* In the ListOps experiments, lower temperatures barely achieved better accuracy.

Decision: Marginally below acceptance threshold
* Improving gradient estimators for discrete latent variable models is an important problem.
* The method is straightforward and the claims of performing better at lower temperatures are supported by empirical evidence.
* However, the overall performance on the ListOps dataset is lower than related work [1], and there does not appear to be a large gain from low temperatures.

Questions:
* The main argument of this paper hinges on the claim that lower temperatures result in lower bias of the gradient estimator. This claim seems reasonable, and is supported by figure 2b. Is there a proof or citation for it, and do we know more? It would be nice to know how variance and bias are traded off, as that would tell us how much (or how little) we could gain from training at lower temperatures.
* Is there an explanation for the difference in performance between the 99% accuracy obtained in Havrylov et. al. 2019 [1] and the performance obtained at low temperatures in this paper?
* How does this method perform versus the estimator proposed in Pervez et. al. [2], which is also single-evaluation?

Suggestions:
* The GR estimator is not guaranteed to have lower variance than ST-GS, just not higher.
* Is there an application where lower temperatures are necessary for training? That would strengthen the argument.

[1] Serhii Havrylov, German Kruszewski, and Armand Joulin. Cooperative Learning of Disjoint Syntax and Semantics. In Proceedings of NAACL 2019.

[2] Pervez, A., Cohen, T., & Gavves, E. 2020. Low Bias Low Variance Gradient Estimates for Hierarchical Boolean Stochastic Networks. ICML 2020.

Edited score after author comments.

---

> ### Author Response · Authors · 2020-11-18
> **MNIST example is a great example of benefits of training at other (lower) temperatures.**
>
> *> “Relative gains from training at lower temperatures are [not] significant”/ “Overall gains over ST-GS seem to be modest on MNIST”/ “There does not appear to be a large gain from low temperatures.”/ “Is there an application where lower temperatures are necessary for training?”*
>
> We politely disagree. We believe our VAE example on MNIST is a great example to demonstrate the benefits of lower temperature training and the improvements are significant:
>
> We achieve up to two nats improvement. To put this into perspective, this is comparable to the improvements of other important innovations in VAE training (e.g., IWAE by Burda et al., 2016 [3]). As R2 points out the improvements are also “very consistent”, and even “competitive with the concrete-relaxation”. Indeed, our results indicate that these improvements are mainly caused by the ability of our estimator to facilitate lower temperature training, because “the best models using ST-GS trained at an average temperature of 0.65, while the best models using GR-MC1000 trained at an average temperature of 0.35”.
>
> Additional References:
> [3] Burda, Y., Grosse, R., & Salakhutdinov, R. 2016. Importance Weighted Autoencoders. ICLR 2016.

---

> ### Author Response · Authors · 2020-11-18
> **We extend the space of trainable temperatures.**
>
> *> [Lower temperatures result in lower bias of the gradient estimator. [...] Proof or citation?]*
>
> We would like to clarify that the contribution of our paper does not hinge on this argument. Still, you make a great point and it has made us reconsider the messaging. We have clarified the following two points throughout the paper.
>
> * The key advantage of our estimator is that it produces variance improvements for all (!) temperatures, thus extending the region of suitable temperatures over which one can tune. This allows a practitioner to explore an expanded set when trading-off of bias and variance.
> * All empirical evidence (including our own) suggests that lower temperatures result in lower bias for the gradient estimator. This has turned into a widely held belief in the community, but as far as we know, it has not been proven. What is known is that the bias in the forward pass of relaxed estimators is reduced: As the temperature is reduced, the relaxed loss converges (see for example Paulus et al. (2020) and references therein) to the true loss. While we are not aware of any work that addresses the convergence of the derivative, it may be possible to use the result in Rudin [1976, Theorem 7.7]. This is an interesting avenue for future work!

---

> ### Author Response · Authors · 2020-11-18
> **We implemented the estimator in [2] and it is outperformed by our estimator.**
>
> *> “How does this method perform [against] the estimator in [2]?*
>
> Thanks, this is a cool paper! Unfortunately, it only works for binary variables, while ours works for all arities. Still, we implemented it as a baseline for our binary VAE and have included the results in a revision: The estimator from [2] improved over ST, but was outperformed by our estimator.

---

> ### Author Response · Authors · 2020-11-18
> **There are significant experimental differences between [1] and us, which we now clearly highlight.**
>
> *> “The overall performance on the ListOps dataset is lower than related work. [...] Is there an explanation for the difference?”*
>
> Thank you for raising this point. Yes, there is an explanation. We were interested in a controlled setting to investigate the influence of K and \tau, while [1] focus on cracking ListOps. These are the most important experimental differences:
> * [1] does not use single-evaluation estimators, we do: They report near perfect accuracy only when using the self-critical baseline. This baseline requires an additional forward pass. All their single-evaluation results are in a similar ballpark as ours accounting for the additional differences below.
> * [1] uses extensive hyperparameter tuning, we do not: They tune learning rate, learning rate schedule, weight decay, entropy regularisation, variance reduction hyperparameters, optimizer (Adadelta), number of updates for PPO, leaf transformations and train for 300 epochs. In contrast, we only tune the (constant) learning rate via gridsearch and use SGD (Appendix D1) for each temperature and train for ten epochs.
> * [1] uses customized training procedures, we are simply plug-in: They use PPO, gradient normalization, different control variates and entropy regularization. We simply plug our estimator into the model from Choi et al. (2018).
> * [1] uses a model with more parameters than us: We do not use any leaf LSTM. It improves performance [Choi et al. (2018)], but may also confound tree learning (e.g. leaf-LSTM may learn to solve the task, making tree obsolete), so we do not use it.
> * [1] uses more training data than us: We reserved 10% of the training set for validation, while they use less than 2%.
>
> We think this is an important discussion, so in our revision, we added a reference to [1] in the main body and highlighted the most important differences in the appendix.

---

> ### Author Response · Authors · 2020-11-18
> **Thank you! We made revisions based on your feedback.**
>
> **Thank you very much for your thoughtful review. We have made revisions to our paper based on your feedback and address your concerns in more detail in the comments below.**

---

> > ### Comment · AnonReviewer1 · 2020-11-18
> > **Great response**
> >
> > Thank you for the thorough response, and correcting my interpretation of the argument. All of my concerns have been addressed, although I do have some lingering concerns about the performance on ListOps in comparison to Havrylov et. al. I am still worried that there is a large gap between the upper bound performance by training via the proposed gradient estimator vs an unbiased one, but that does not detract from the contribution of the paper which demonstrates improvements over the ST estimator baseline. There are enough confounding factors, in particular model differences, that it is unclear whether the results in Havrylov et. al. are comparable. I will update my score accordingly.

---

### Author Response · Authors · 2020-11-18
**We made revisions based on the feedback of the reviewers.**

**We have incorporated the feedback of the reviewers and summarize here the changes in our revision (new upload!):**

* Additional baselines (R1, R4): We ran experiments for FouST (Pervez et al., 2020, as kindly suggested by R1) where applicable (binary VAE): FouST improved clearly over ST, but is outperformed by our method.
* Low temperatures and low bias (R1): We have clarified throughout the paper that our method extends the space of trainable temperatures to those that have lower bias, but higher variance. Empirically, these tend to be low temperatures.
* ListOps experimental set-up (R1): We added a reference to Havrylov et al. (2019) in the main body and highlight the differences between their and our experimental set-up in the appendix to put results into perspective.
* We clarified the discussion in 3.3 as suggested by R2 and added a footnote.

Please see individual responses below for more details.

---

### Decision · Program_Chairs · 2021-01-07
**Final Decision**

**Decision:**

Accept (Oral)

**Comment:**

The paper presents a variance reduction technique to the Straight-Through version of the Gumbel-Softmax estimator. The technique is relying on the truncated Gumbel of Maddison et al. I share the excitement of the reviewers about this work and I expect this technique to further influence the field.